# Development of Green Banana Fruit Wines: Chemical Compositions and In Vitro Antioxidative Activities

**DOI:** 10.3390/antiox12010093

**Published:** 2022-12-30

**Authors:** Zhichun Li, Cuina Qin, Xuemei He, Bojie Chen, Jie Tang, Guoming Liu, Li Li, Ying Yang, Dongqing Ye, Jiemin Li, Dongning Ling, Changbao Li, Hock Eng Khoo, Jian Sun

**Affiliations:** 1Agro-Food Science and Technology Research Institute, Guangxi Academy of Agricultural Sciences, Nanning 530007, China; 2Guangxi Key Laboratory of Fruits and Vegetables Storage-Processing Technology, Nanning 530007, China; 3College of Chemistry and Bioengineering, Guilin University of Technology, Guilin 541006, China

**Keywords:** active ingredient, antioxidant activity, aroma components, banana peel, chemical composition

## Abstract

This study aimed to develop functional fruit wines using whole fruit, pulp, and peels from green bananas. The boiled banana homogenates were mixed with cane sugar before wine fermentation. Quality parameters, phenolic compounds, flavor components, and antioxidative properties of the green banana peel wine (GBPW), green banana pulp wine (GBMW), and whole banana wine (GBW) were determined. High-performance liquid chromatography was used to determine the phytochemical compounds in three wines, and the flavor components were further analyzed using headspace solid-phase microextraction combined with gas chromatography–mass spectrometry. The flavor components and in vitro antioxidant activities were, respectively, determined using the relative odor activity value and the orthogonal projections on latent structure discrimination analysis (OPLS-DA). In vitro antioxidative capacities for these wines were evaluated using antioxidant chemical assays and cell culture methods. The total phenolic and total tannin content of the GBPW, GBMW, and GBW showed reducing trends with increasing fermentation days, whereas the total flavonoid content of the wine samples exhibited downward trends. The antioxidant capacities of the three wine samples were higher than those of the raw fruit samples, except for the metal chelation rate (%). Additionally, the main flavor component in the wine samples was 3-methyl-1-butanol. Its percentages in the GBPW, GBMW, and GBW were 72.02%, 54.04%, and 76.49%, respectively. The OPLS-DA results indicated that the three wines presented significantly different antioxidant activities. The cell-culture-based antioxidant analysis showed that these wine samples had protective effects against the oxidative stress of the 3T3-L1 preadipocytes induced by hydrogen peroxide. This study provided a theoretical basis for defining the antioxidant characteristics of banana wines and expanding novel channels for using banana peels to develop nutraceuticals.

## 1. Introduction

Bananas are one of the most cultivated and consumed fruits in the world. They are also one of the crucial fruit varieties grown in South China, with an average annual output of 11,130.02 million tons from 2016 to 2020, accounting for 27.23% of the total fruit output. The skin or peel accounts for about 40% of the total fruit weight of a ripe banana. Banana processing often generates many by-products, such as peels, which are rarely eaten directly owing to their bitter taste. Banana peels are by-products or waste, which poses serious environmental hazards [1]. Due to the mass production of banana peels from banana product processing, the fruit peel has been used as compost or produced into more valuable industrial products. Additionally, a high percentage of postharvest green bananas present black spots on the skin. These bananas have a low commercial value and are not marketable. They are commonly discarded as wastes and also pollute the environment.

Bananas have been developed into different functional foods to date. Although banana peels are not commonly used as food, they are typically used as a carbon-based adsorbent for wastewater treatment [2], fuel [3], and animal feed [4]. Previous literature showed that green banana powder contained a high amount of resistant starch, which can be used as a nutritional or functional component [5]. Bananas have high nutritional and medical values due to their colorful pigments and high carbohydrate content [6]. The starch obtained from bananas has high economic values as food and medicine [7]. In banana production and processing, tonnes of residual fruits, tail fruits, and peels are wasted without specific recycling treatment and particular utilization.

Studies on green bananas can be divided into fundamental research and product development, of which the product characteristic determination includes physicochemical and digestive properties [8,9]. The application of bananas is adding banana-based starch as a functional component in developing food products [10,11]. These studies are significant for the processing and use of bananas and their by-products. In addition, the previous reports on banana peels mainly focused on the analyses of physicochemical characteristics and the application of bioactive compounds such as phenolic compounds [12], dietary fibers [13], pectins [14], and tannins [15]. These bioactive compounds have anticancer [16], antibacterial [17,18], antidiabetes [19], and antioxidant capacities [20], and they prevent cardiovascular diseases [21,22]. For example, Durgadevi et al. [23] reported the effect of banana peel samples on the proliferation of human breast cancer cells cultured in vitro.

A large volume of by-products is produced annually during banana production and processing because no special recycling treatment is available. Therefore, fully utilizing these by-products to improve the economic benefits of the banana industry is necessary and urgent. Applying green banana peels in wine making can reduce waste production and increase resource utilization. Furthermore, determining the flavor components and antioxidative capabilities of green banana wines is also helpful for functional product development and recycling banana by-products or alternative uses.

## 2. Materials and Methods

### 2.1. Sample Preparation

The unripe green bananas (*Musa acuminate* L.) of the ‘Guijiao 9’ variety were harvested from the banana germplasm resource nursery of Jinghong city, Xishuangbanna, Yunnan, in April 2021. Green bananas were obtained during the immature stage, packed, and transported to a wine fermentation laboratory. One kilogram of each whole fruit, pulp, and peel was prepared, pulverized, and homogenized with 3.5 L of distilled water. Then, 500 mL of each sample was separately collected from the 3.5 L mixture as a fresh fruit sample. The leftover 3.0 L was boiled for 10 min to denature all the enzymes and destroy the microbes. The hot banana samples were mixed with sugar (136 g/L), homogenized, and cooled to 40 °C before adding 0.5 g/L pectinase and 0.3 g/L cellulase. Then, the mixture was hydrolyzed for 2 h and added with 0.02% dried wine yeast K1-V1116 (*Saccharomyces cerevisiae*). The wine fermentation was conducted at room temperature, and test samples were collected at 0, 3, 6, 9, 15, and 30 days of fermentation. The sterilization was performed by heating the fermentation medium to 65 °C and maintaining it for 15 min.

### 2.2. Determination of Quality Parameters and Bioactive Components

Banana wine samples were filtrated before further analysis. The selected quality parameters, including the total acid content (TAC), pH value, total soluble solids (TSS) content, and alcohol content, were determined according to the methods described by Li et al. [24]. The total phenolic content (TPC) and total tannin content (TTC) were assayed using the Folin–Ciocalteu method, whereas the total flavonoid content (TFC) was determined using the aluminum nitrate method [25]. The TPC and TTC were expressed as the gallic acid equivalent (mg/g extract). The TFC was expressed as the quercetin equivalent (mg/g extract).

### 2.3. UHPLC-MS Identification and Quantification of Phenolic Compounds

The phenolic compounds in the banana wine samples were determined using ultrahigh pressure liquid chromatography (UHPLC) coupled with a mass spectrometer (MS) on an Acquity UPLC HSS T3 column (2.1 mm × 100 mm, 1.7 μm, Waters, Milford, MA, USA). The mobile phase A was 0.1% aqueous formic acid, and the mobile phase B was acetonitrile [26]. The flow rate was 0.25 mL/min. The gradient elution procedure was 0–0.8 min, 90% A; 0.8–4.0 min, 90–80% A; 4.0–6.5 min, 80% A; 6.5–7.0 min, 80–55% A; 7.0–13.0 min, 55% A; 13.0–13.5 min, 55–90% A; and 13.5–14.0 min, 90% A. The column temperature was set at 35 °C, and the injection volume was 2 μL. MS scanning was based on electrospray ionization sources (ES+ and ES−), scanning mode multireaction monitoring (MRM), a capillary voltage of 3.0 kV, and an ion source temperature of 350 °C. The solvation gas was nitrogen, with a flow rate of 700 L/h. The flow rate of the cone-hole gas (nitrogen) was 150 L/h, and the impact gas flow rate (Ar) was 0.10 mL/min. The phenolic compounds in the wine samples were analyzed using the 18 phenolic standards. Five different concentrations (0.004–8.0 mg/mL) were selected to plot the standard calibration curves. These standard calibration curves had a very high linearity (>0.99). The screening test was performed using a UV-DAD detector at a wavelength of 280 nm.

### 2.4. HS-SPME-GC-MS Analysis

The flavor components in the banana wine samples were analyzed using a headspace solid-phase microextraction combined with gas chromatography–mass spectrometry (HS-SPME-GC-MS, Agilent Technologies, Santa Clara, CA, USA) [27]. The GC column was HP-5 MS (30 m × 250 μm × 0.25 μm). The initial column temperature was set to 40 °C and was maintained for 5 min; then, it was increased to 120 °C at 4 °C/min rate and was let to stand for 3 min; then, it was increased to 220 °C at 10 °C/min and was held for 10 min; finally, it was increased to 280 °C at 30 °C/min and was maintained for 15 min. The carrier gas was high-purity helium with a flow rate of 1 mL/min. The inlet temperature was 250 °C. The MS conditions were an EI ion source, electron energy of 70 eV, electron multiplier voltage of 1376 V, scanning range from 15 to 500 amu, ion source temperature of 230 °C, and four-stage rod temperature of 150 °C. The flavor compounds were identified via spectra matching using the GC/MS library. Their ion masses were also compared with the data reported in online databases (PubChem and other databases, accessed on 20 June 2021).

The method for relative odor activity value (ROAV) was used to quantify the flavor components in the banana wine samples. The odor activity value (OAV) was calculated according to Equation (1):(1)$$\mathrm{OAV}=\frac{C}{T}$$
where C is the relative concentration of the flavor substance and T is the sensory threshold of this flavor substance.

The ROAV_max_, with the highest contribution rate, was marked as 100. The ROAV values of the other remaining aroma components were calculated according to Equation (2):(2)$$ROAV=\frac{C_{i}}{C_{S}\tan}\times \frac{T_{S}\tan}{T_{i}}\times 100$$

*C_i_* is the relative content of the aroma components, *T_i_* is the sensory threshold of the aroma components, and *C_S_* tan and *T_S_* tan were the relative content and sensory threshold of the aroma components contributing the most, respectively.

The sensory threshold of the flavor components was determined by Shi et al. [28]. According to the calculated result (Equation (2)), a ROAV ≥ 1 denoted that the substance directly impacted the overall flavor components and was a principal compound. If 0.1 ≤ ROAV < 1, it indicated that the flavor substance possessed a modification effect on the total flavor substance and was a modified compound. If the ROAV < 0.1, it showed that the flavor substance had no significant influence on the overall flavor.

### 2.5. Determination of In Vitro Antioxidant Activity

The antioxidant activities of the banana wine samples were determined using the DPPH radical scavenging rate, hydroxyl radical scavenging ability, superoxide anion radical scavenging ability, and metal chelation rate. The DPPH radical scavenging rate was measured using the DPPH free radical scavenging assay [29], and the metal chelation rate was determined using an iron (II) chloride solution coupled with the ferrozine coloration method [30]. The hydroxyl radical and superoxide anion radical scavenging abilities were also determined with respective test kits according to the manufacturers’ instructions. Vitamin C (Vc) was used as a standard for comparison. A metal chelation assay is also known as a FRAP assay.

### 2.6. Cell-Culture-Based Antioxidant Capacities and Proliferation Inhibition Assays

The antioxidant capacities and inhibition abilities of the banana wine samples were determined using the oxidative stress model of 3T3-L1 preadipocytes induced by 30% hydrogen peroxide. A cell-culture-based oxidative stress model was developed and established by Kim et al. [31], and the antioxidant capacities of the samples were tested using a previously established method [32]. Healthy preadipocytes were treated with 20 μL of the banana wine samples for 48 h with or without adding 20 μL of 30% hydrogen peroxide. The negative control of the cell-culture-based antioxidant capacity was the cells without adding banana wine samples and without being induced by the 30% hydrogen peroxide. The positive control was the cells only induced by hydrogen peroxide. The cell viability was analyzed with an MTT assay.

### 2.7. Statistical Analysis

All the data are presented as the means of three replicates. The statistically significant differences in the antioxidant activities between the samples were determined using the analysis of variance (ANOVA) coupled with a posthoc LSD test. SPSS version 24 (SPSS Inc., Chicago, IL, USA) was used to analyze the mean differences at a *p*-value less than 0.05. The antioxidant activities of the three green banana wine samples, determined using four in vitro antioxidant activity assays, were analyzed with OPLS-DA.

## 3. Results

### 3.1. Quality Parameters and Bioactive Components

The TSS, alcohol content, pH value, and TAC are important quality indexes of fermented fruit wines. These quality parameters are closely related to the taste and flavor of wines. These quality parameter values for the GBPW, GBMW, and GBW samples are shown in Figure 1. The result showed that the TSS in the fruit wine samples decreased with an increasing fermentation time, with a remarkable reduction observed within six days. The alcohol content increased the most during this fermentation period and then elevated more gradually (Figure 1A). After day 6, the decline in the TSS was less than 1%. The reduction rate remained similar until day 30 of the wine fermentation. The alcohol content in the GBW sample started to reduce on day 9 of fermentation. The average reduction rates of the TSS in the GBPW, GBMW, and GBW samples on day 3 of fermentation were 44.5%, 50.0%, and 46.1%, respectively, while the alcohol contents increased to 3.5%, 4.9%, and 4.5%, respectively.

After the third day of fermentation, the increment of the alcohol content and the reduction in the TSS became slower, probably because the yeast multiplied rapidly at the early fermentation stage, and the metabolism rate increased. It digested soluble solids in a fermentation solution to provide sufficient nutrients and convert them into alcohol, organic acids, and other flavor substances. At a later fermentation period, soluble solids in the wine samples were lower. The low amount of soluble carbohydrates could not provide enough energy for the growth and survival of the yeast. Therefore, yeast entered the aging period and finally lyzed. The yeast produced increasing amounts of alcohol and organic acids during fermentation. Additionally, the GBMW showed the most notable decreasing TSS level. The TSS in the GBPW was lower than the other wine samples, and the alcohol content in the GBPW increased the least.

The changes in the pH value and TAC of the fruit wine samples are depicted in Figure 1B. The TAC in the wine samples had increasing rates, whereas the pH value showed decreasing values. The increasing and decreasing rates of the TAC in these wine samples varied at different fermentation stages. From day 0 to day 3, the TAC in the GBPW, GBMW, and GBW increased to 13.5%, 19.9%, and 18.1%, while the pH values decreased to 29.0%, 27.2%, and 27.3%, respectively. After the third day of fermentation, the total acid content in these wine samples had a slower increment rate and finally decreased, except for the GBMW. The pH values started to increase on day 3 of fermentation, but the increment rate was slow until day 30.

At the early fermentation stage, the TAC drastically increased because the yeast rapidly multiplied, and a high metabolic rate produced a higher organic acid content. It caused the pH value of the sample to rapidly decrease to a pH of 4. Due to the influences of a low pH value, substrate concentration, and other related factors, yeast metabolism slowed down at the later stage of fermentation. The production rate of the organic acids also decreased; it was easier for the acids to undergo an esterification reaction with alcohols, and the degradation of organic acids occurred and produced ethanol. The reduced concentration of the organic acids led to a lesser increase in the pH value. During wine fermentation, the GBMW had the most notable increase in the TAC, but the pH values of the three fruit wines showed no remarkable differences.

The TPC, TTC, and TFC in the GBPW, GBMW, and GBW during fermentation are shown in Figure 2. The result showed that the TPC in the GBMW and GBW showed overall decreasing values during fermentation (Figure 2A). During 0–3 days of fermentation, the TPC decreased sharply from 15.10 mg/g and 16.71 mg/g to 3.50 mg/g and 9.44 mg/g, respectively. After the third day, the GBW decreased slowly until the end of fermentation, while the GBMW increased with a minor fluctuation. The TPC in the GBPW increased from 20.15 mg/g to 25.28 mg/g from baseline to the sixth day of fermentation and then decreased to 20.93 mg/g on day 9. The TPC finally increased slightly until the end of fermentation. The TPC of the GBPW was significantly higher on day 30 of fermentation than on day 0 (*p* < 0.01). The increased TPC during the early stage of fermentation could be due to the digestion of carbohydrates, thus releasing free phenolic compounds.

The reduction in the TPC of the GBMW and GBW could be due to the increased concentration of the soluble polysaccharides in the extracts. Similarly, the TPC in the GBPW was reduced by about 5 mg/g. This could have been due to the same reason. The further digestion and utilization of banana starch released more free polyphenolic compounds, except for the GBW. The GBW showed a slight decrease in the TPC from day 15 to day 20 of fermentation. It showed that the minor degradation of polyphenols occurred during fermentation at a temperature of 40 °C. The reduced TPC might also be attributed to the binding and adsorption of secondary metabolites produced by microbial fermentation with polyphenolic compounds or the precipitation and oxidation of polyphenols during fermentation [33].

Figure 2B depicts the TTC in the banana wine samples. During the fermentation process, the TTC in the GBMW and GBW showed decreasing rates. A rapid reduction was noted from day 0 to day 3 of fermentation, where the value reduced from 4.81 mg/g and 3.43 mg/g to 1.41 mg/g and 2.30 mg/g, respectively. The decreasing rate was slower after day 3 until the end of fermentation. The TTC in the GBMW and GBW at the end of fermentation was 1.30 mg/g and 1.32 mg/g, respectively. The TTC in the GBPW increased from 5.02 mg/g on day 0 to 6.17 mg/g on day 3 of fermentation and then reduced to 5.56 mg/g on day 6. A slight reduction in the TTC was observed from day 15 until the end of fermentation. The TTC in the GBPW on day 30 of fermentation was significantly higher than on day 0 (*p* < 0.01). The TTC in the GBPW increased on the first three days of fermentation due to the rapid digestion of complex tannin in the banana peel. The reduced TTC could be due to the polymerization reaction of tannins. This result showed that the TTC in the GBPW was significantly higher than in the GBMW and GBW. It could be related to a higher TTC in the green banana peel. Additionally, changes in the TPC and TTC of the fruit wine samples had similar trends.

Figure 2C shows the changes in the TFC in the banana wine samples. The TFC in the GBPW, GBMW, and GBW increased during the first few days of fermentation, then gradually reduced until the end of the fermentation process, except for the GBPW. The TFC in the GBPW, GBMW, and GBW increased from 11.55, 2.69, and 3.20 mg/g on day 0 to 35.10, 4.74, and 6.68 mg/g on day 3, respectively. The TFC in the GBPW rapidly increased until day 9 and then reduced until day 15. It had a minor increase from day 15 until the end of fermentation. The variation in the TFC during fermentation could be because flavonoids are unstable compounds. They were easily affected by changes in the solution pH, reducing agents, light, and other factors. During the fermentation process, the TFC in the GBPW was significantly higher than in the GBMW and GBW. It could be related to the flavonoid content in the banana peel, which was higher than the fresh pulp [34].

### 3.2. Identification and Quantification of Phenolic Compounds

The mass spectrometric parameters of 18 phenolic standards are shown in Table 1. The concentrations of these phenolic compounds in the GBPW, GBMW, and GBW were determined and are shown in Table 2. Five phenolic acids were identified from three fermented wine samples. They were salicylic, chlorogenic, ferulic, and p-coumaric acids. Among them, salicylic acid accounted for the highest proportion in the GBPW and GBMW, whereas the GBW had the highest concentration of ferulic acid. The ferulic acid concentration in the GBPW and the salicylic acid concentration in the GBMW were significantly higher than in the other two wine samples. Catechin was detected only in the GBPW, and chlorogenic acid was not found in the GBW. Salicylic acid was detected in the banana-peel-based wines because this compound was used for the postharvest treatment of bananas to extend their shelf life [35].

Apigenin, coumarin, and luteolin are flavonoids that were not detectable in these wine samples. Kaempferol was the only flavonoid detected in all the banana wine samples, where the GBPW exhibited the highest concentration. Additionally, the initial screening of the phenolic compounds was performed using the UHPLC method coupled with a UV-DAD detector. The results showed that gallic acid was not detected in the banana wine samples. The differences in the types and phenolic compound concentrations of the banana wine samples might be related to plant origin, microbial species, and several fermentation parameters [36].

### 3.3. HS-SPME-GC-MS Analysis of Flavor Components

The relative contents of the flavor components in the GBPW, GBMW, and GBW are shown in Table 3. A total of 22 flavor substances were identified from the GBPW, GBMW, and GBW. They consisted of four alcohols, thirteen esters, one acid, three alkenes, and one phenol. The relative contents of five different types of aroma substances in three wine samples are shown in Figure 3. The result showed that alcoholic compounds were the main flavor substances in the GBPW, GBMW, and GBW. Among the flavor components, 3-methyl-1-butanol was the highest in the GBPW, GBMW, and GBW, with levels of 72.02%, 54.04%, and 76.49% of the total flavor content, respectively. This alcohol compound has unique grass and a mature fruit aroma [37]. Fatty acid esters in the GBMW were more abundant than in the other two wine samples. Ethyl palmitate, 3-methyl-1-butanol, and 2,4-di-tert-butylphenol were detected in the GBPW, GBMW, and GBW. 2,3-Butanediol was found only in the GBPW and GBMW but not in the wine sample produced from whole bananas. On the contrary, alkenes and acetic acid were not detected in the banana peel wine samples.

The ROAV values of the flavor components in the three wine samples are shown in Table 4. The key flavor component in the wine samples was 3-methyl-1-butanol. Ethyl palmitate was another contributor to the aroma of the wines. The flavor of the GBMW was mainly due to 3-methyl-1-butanol and ethyl octanoate. Ethyl palmitate, ethyl caprate, ethyl laurate, ethyl undecanoate, ethyl myristate, and methyl palmitate also enhanced the flavor of the GBMW. Methyl palmitate was another potent flavor compound in the wine sample produced from the banana pulp. The main flavor components in the GBW were ethyl palmitate, ethyl laurate, and ethyl myristate, in addition to 3-methyl-1-butanol. The other flavor components did not contribute to the aroma of these banana wines.

The flavor component is an essential index of fruit wine quality. The sensory quality of a fruit wine is attributed to the volatile and aromatic compounds in the wine. These flavor components indirectly affect the commodity value of the fruit wine. The flavor substances in a fruit wine mainly consist of alcohols, esters, acids, and alkenes, which are related to fruit aroma. Among them, alcohols and acids have relatively high flavor thresholds and contribute little to the flavor of fruit wine. On the other hand, esters have relatively low flavor thresholds and thus contribute more to the aroma of fruit wine.

### 3.4. In Vitro Antioxidant Capacities

The DPPH radical scavenging activity, hydroxyl radical scavenging ability, superoxide anion radical scavenging ability, and metal chelation rate of the GBPW, GBMW, and GBW are shown in Figure 4. The results showed that different banana parts used in wine fermentation affected DPPH radical scavenging activities (Figure 4A). The scavenging activities were statistically different between the banana fruit and banana wine samples. The DPPH radical scavenging activities of the wine samples were remarkably higher than that of the green banana samples (*p* < 0.01). The green banana peel (GBP) had the significantly lowest DPPH radical scavenging activity (*p* < 0.05), whereas the GBMW had the highest activity. We also found no significant differences between the DPPH radical scavenging activities of green banana pulp (GBM) and whole green banana (GB) (*p* > 0.05). Moreover, no statistical differences were observed between the scavenging activities of Vc and green banana wine samples (*p* > 0.05).

The hydroxyl radical scavenging abilities of the banana wine samples are depicted in Figure 4B. The fermentation conditions and different parts of the green banana had a remarkable impact on the hydroxyl radical scavenging ability of the green banana wine samples. The hydroxyl radical scavenging abilities of the banana wine samples were significantly higher than those of the green banana samples (*p* < 0.01). The scavenging ability of the GBPW was also statistically the highest among the banana fruit parts and wine samples (*p* < 0.01). Moreover, hydroxyl radical scavenging ability of Vc was not as high as the green banana wines. Moreover, significant differences were observed between the scavenging abilities of the different green banana wine samples.

Based on the data depicted in Figure 4C, wine fermentation significantly affected the superoxide anion radical scavenging ability. The scavenging abilities of the green banana wines were remarkably higher than those of the green banana fruit samples (*p* < 0.01). The GBPW had the significantly highest scavenging ability (*p* < 0.05). We found no significant differences between the scavenging abilities of the different fruit parts (*p* > 0.05). On the contrary, the scavenging ability of Vc was significantly the highest among all the samples (*p* < 0.05).

The metal chelation rates of the fruit samples were remarkably lower after the wine fermentation (Figure 4D). The metal chelation rates of the green banana wine samples were over three times lower than those of the green banana samples. The GBMW had the significantly lowest metal chelation rate (*p* < 0.05), indicating that the banana pulp had a lower metal chelation rate than the peel. This finding revealed that the fermentation conditions and types of raw materials altered the metal chelation rate of the fermented wine. The metal chelation rate of the green banana fermented wine was significantly lower than that of its raw material (*p* < 0.01), whereas the metal chelation rates of the green banana skin, pulp, and fruit samples were significantly different from each other (*p* < 0.01).

### 3.5. Comparison of Antioxidant Activities of Wine Samples Based on OPLS-DA

OPLS-DA data are depicted in Figure 5. The total variance of the samples was 99.02%, R2 was 82.3%, and Q2 was 72.2%. Three groups of samples were divided into four quadrants (Figure 5A), with the GBPW located in the first and fourth quadrants, the GBW in the second quadrant, and the GBMW in the third quadrant. A permutation test was subsequently performed to verify the model by randomly rearranging the experiments by changing the ranking order of the categorical variables (Y) and randomly assigning a Q2 of up to 200 times. Figure 5B shows the results of the permutation test. The intersection between the regression line and the vertical axis of point Q2 was lower than zero. It indicated that the discriminant model was not overfitted. Therefore, the initial model outperformed the random permutation model. An antioxidant activity landmark of different wine samples was selected based on the variable influence on projection (VIP), with a VIP value higher than one, namely superoxide anion radical scavenging abilities (Figure 5C). The results showed that the superoxide anion radical scavenging assay was an important antioxidant assay for discriminating the antioxidant activity of these wine samples.

### 3.6. Cell-Culture-Based Antioxidant Capacities and Cell Viability of 3T3-L1 Preadipocytes

The antioxidant capacities of the green banana wine samples were determined by an inhibition assay of hydrogen-peroxide-induced oxidative stress on 3T3-L1 preadipocytes. The antioxidant capacities of these wine samples are shown in Figure 6A. The antioxidant capacities of the wine samples were higher with the increase in sample concentrations. At sample concentrations from 0 to 25 mg/mL, the antioxidant capacities of the green banana wine samples were lesser than 50%. Only the GBPW at 25 mg/mL had an antioxidant capacity higher than 50%. At sample concentrations from 50 to 100 mg/mL, all the wine samples had higher antioxidant capacities than 50%, except for the GBW at a concentration of 50 mg/mL. Additionally, the antioxidant capacity of the GBW was significantly the lowest at a sample concentration of 100 mg/mL. The GBMW at lower concentrations had a significantly lower antioxidant capacity than the GBMW and GBW. At higher sample concentrations, the GBMW had an antioxidant capacity comparable to the GBPW. The results showed that the three wine samples could protect 3T3-L1 preadipocytes against oxidative stress induced by hydrogen peroxide. It could be because the GBPW, GBMW, and GBW contained polyphenolic and other compounds with moderate to high antioxidant activities, which could remove free radicals [38].

The protective effects of the three wine samples against hydrogen-peroxide-induced oxidation to 3T3-L1 preadipocytes are shown in Figure 6B. The IC_50_ value of the GBPW was significantly the lowest, followed by the GBW and GBMW. The highest IC_50_ value of the GBMW indicated that it had the least inhibitory effect on the growth of 3T3-L1 preadipocytes. The results showed that the wine produced from the green banana peel had the highest protective effect against cellular oxidative stress compared to the fruit pulp-based wines. Additionally, the in vivo antioxidant capacity of the GBPW was the highest at sample concentrations between 25 and 100 mg/mL. In this study, 3T3-L1 preadipocytes could be specifically induced to differentiate into mature adipocytes [39]. Therefore, it was best to represent a cellular system for an antioxidative assay.

## 4. Discussion

Green bananas are rich in polyphenolic and other compounds with high antioxidant properties. Green banana waste, such as banana peels, has been used as a raw material to produce functional materials. Banana peels are prebiotics and biosubstrates [22]. They contain biosubstrates, such as pectin and cellulose. The polysaccharides extracted from banana peels can provide energy for microorganisms. Therefore, banana peels have been used to produce alcohol in the past. The use of banana peels in wine production enhanced the antioxidant activity of the fruit wine because the banana peel is high in antioxidants [20]. The ferric-reducing ability of the banana peel was also higher than the fruit pulp. The high antioxidant activities of the banana wine samples could be attributed to their high levels of bioactive antioxidants. Additionally, the findings of this study revealed that the GBPW had remarkably high phenolic compounds, and the antioxidant activities of the wine samples were also higher than the banana fruit samples. Moreover, the DPPH radical scavenging activity of the green banana peel reported in a previous study was remarkably higher than that of a yellow banana peel [40].

The quality of green banana wines was influenced by the other chemical properties in addition to their bioactive antioxidants. The GBPW had a remarkably low alcohol content compared with the other two wine samples. The results showed that wine fermentation using a banana peel had a low alcohol content in the wine sample. The possible reason could be that it had a carbohydrate content lower than the banana pulp. A previous study supported our findings that the acidity and TSS of the GBMW reduced gradually during the 30 days of fermentation [41]. The TSS of the GBPW was notably lower than that of the GBMW. The GBMW also had higher acidity than the GBPW. Therefore, the GBPW had a better quality than the GBMW. Additionally, the green banana wine samples had a lower alcohol content (<10%) compared with the alcohol content of ripe banana pulp wine (15.49%) reported in the literature [42]. It could be due to the ripe banana pulp having a higher sugar content. The alcohol produced during the banana wine fermentation could also be due to the fermentation time, where the banana wine produced from 14-day fermentation had 5% alcohol [42].

Different parts of bananas, such as the peel, pulp, and whole fruit, contain different percentages of phenolic compounds and flavor components. This study compared the fruit wines produced from three parts of bananas because banana pulp wine has a better flavor, but its antioxidant activity could be lower than banana peel wine. The use of the whole banana in wine production somehow improved its aroma and antioxidant activity. The banana peel wine (GBPW) had higher antioxidants than its pulp wine, but the whole banana fermented wine (GBW) had a lower IC_50_ value than the GBMW. The production of fruit wine using bananas with the peel attached maintained a high antioxidant level in addition to improving the wine quality. Therefore, the study on developing banana wines by applying different parts of green bananas warrants the wine quality. This study may also provide a theoretical basis for expanding the channel of banana industrialization, increasing the economic value of bananas, and reducing the waste of resources. The reason for using green bananas was that green banana pulp had higher phenolic content than ripe yellow bananas [43].

The aroma of the GBPW was unique because the main flavor component of the wine produced from banana peels was 3-methyl-1-butanol. The literature reported that isoamyl butyrate was the main volatile in banana peels, but the level of its isoamyl alcohol, such as 3-methyl-1-butanol, was low [44]. The high 3-methyl-1-butanol in the GBPW could be due to the conversion of isoamyl butyrate to isoamyl alcohol during fermentation. Isoamyl butyrate has an intense fruity odor, but the metabolism of this compound to isoamyl butanol during fermentation could change its aroma. This flavor substance has a fusel odor. These wine samples had a banana-like smell, which could be due to the butan-1-ol component. The wine odors of the GBMW and GBW were pleasant because these wine samples had floral and fruity aromas from other volatiles.

We identified 22 flavor substances in the GBMW. Most of these substances were fatty acid esters. This study was preliminary work on developing green banana wines using different parts of green bananas. A few other physicochemical parameters and analyses of the wine samples have not been included in this study. Although we have not performed a sensory evaluation for these wine samples, we noted that each green banana wine sample had a unique color and appearance. The GBPW had a yellowish brown hue, the GBMW had a milky yellowish appearance, and the GBW was a pale golden-yellow-colored liquid. Among the three wine samples, the GBPW had the most pleasant odor. It had a fresh and fruity smell and taste due to its higher percentages of 2,4-di-tert-butylphenol and 3-methyl-1-butanol. Only the banana-pulp-based fermented wines had a pungent smell due to the acetic acid in these wines.

The production of the GBMW is more costly because the banana pulp has a higher commercial value than its peel. The antioxidant quality of the GBMW was also lower than the GBPW, although the GBMW had a better aroma. Since this study did not cover sensory evaluation and some other in vitro bioassays for the wine samples such as the GBPW, they should be included in future studies. The GBPW is one of the potent functional alcoholic beverages with high antioxidants for promoting a healthy drinking lifestyle. The wine sample can be used as an alcoholic tonic drink to prevent metabolic diseases among the population with unhealthy lifestyles because salicylic acid is its key compound.

## 5. Conclusions

The chemical compositions and bioactive phenolics of the GBPW, GBMW, and GBW had remarkable changes during the first few days of fermentation. The changes in these values were minor during the following days until the end of fermentation. During the fermentation process, the TSS content of the GBPW, GBMW, and GBW showed reducing levels with increasing fermentation time, whereas the alcohol and TAC had increasing rates but with some exceptions. The alcohol and TAC of the GBW had a downward trend. The TPC and TTC of the GBMW and GBW were decreased during the fermentation. However, the changes were not notable after day 10 of fermentation. The GBPW had the highest TPC, TTC, and TFC compared to the GBMW and GBW. This shows that adding a banana peel to the sugar fermentation improved the quality of the fermented wine, such as by further reducing its alcohol and TSS content and increasing its total phenolic components a few times higher than the banana-pulp-based wine.

These wine samples had higher antioxidant activities than the different banana parts, except the wine samples had a lower metal chelation rate than the fruit samples. The OPLS-DA revealed that the antioxidant activities of the green banana wine samples were notably different, and the antioxidant activity assay that discriminated based on the antioxidant activities of these wine samples was the superoxide anion radical scavenging assay. The three fermented wine samples had a protective effect against oxidative stress induced by hydrogen peroxide in 3T3-L1 preadipocytes. The GBPW had a better inhibitory effect than the other two wine samples at the same concentrations. Based on these findings, adding different banana parts with sugar for wine fermentation improved the wine quality. The addition of the whole banana in wine fermentation is recommended because the full utilization of the banana peel reduces waste production, the banana peel is rich in antioxidants, and banana pulp has a more favorable aroma. Wine is good for preventing cardiovascular diseases among heavy drinkers. Future studies can consider performing a large-scale sensory evaluation of these green banana wines and their physicochemical characteristics.

## Figures and Tables

**Figure 1 antioxidants-12-00093-f001:**
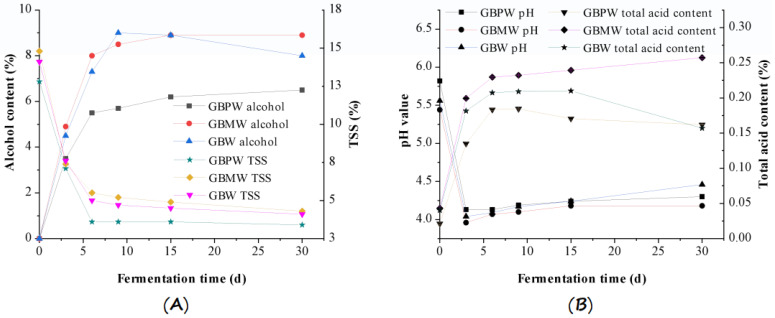
Changes in quality parameters of GBPW, GBMW, and GBW during fermentation of green banana samples. (**A**) Alcohol content (%) and total soluble solids (TSS %); (**B**) pH value and total acid content (%).

**Figure 2 antioxidants-12-00093-f002:**
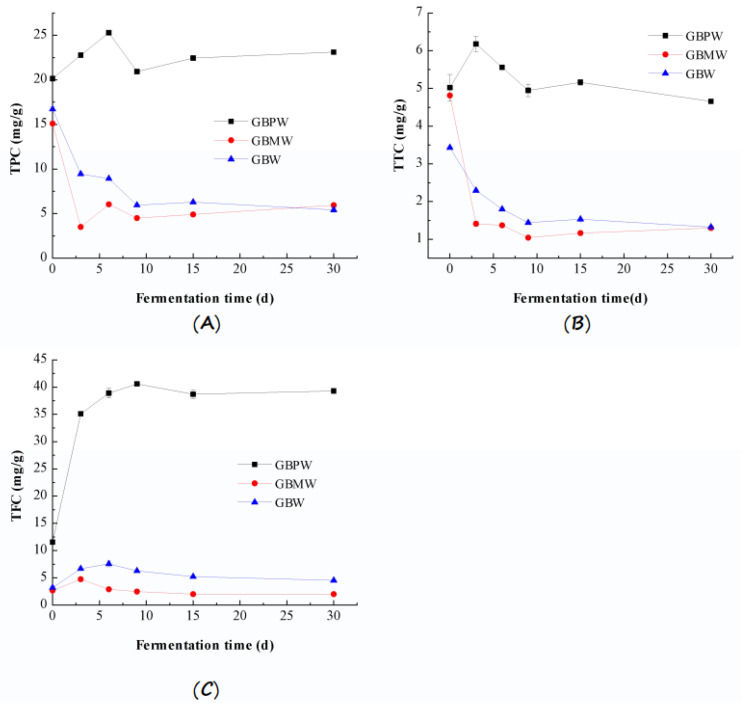
Changes in bioactive components of GBPW, GBMW, and GBW during fermentation of green banana samples. (**A**) Total phenolic content (TPC); (**B**) total tannin content (TTC); (**C**) total flavonoid content (TFC).

**Figure 3 antioxidants-12-00093-f003:**
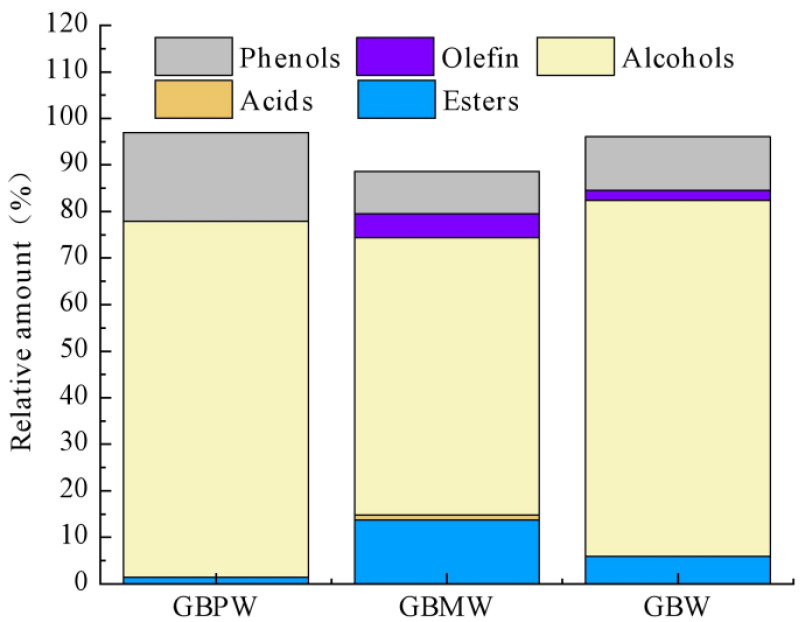
Relative amounts of flavor components in green banana wine samples.

**Figure 4 antioxidants-12-00093-f004:**
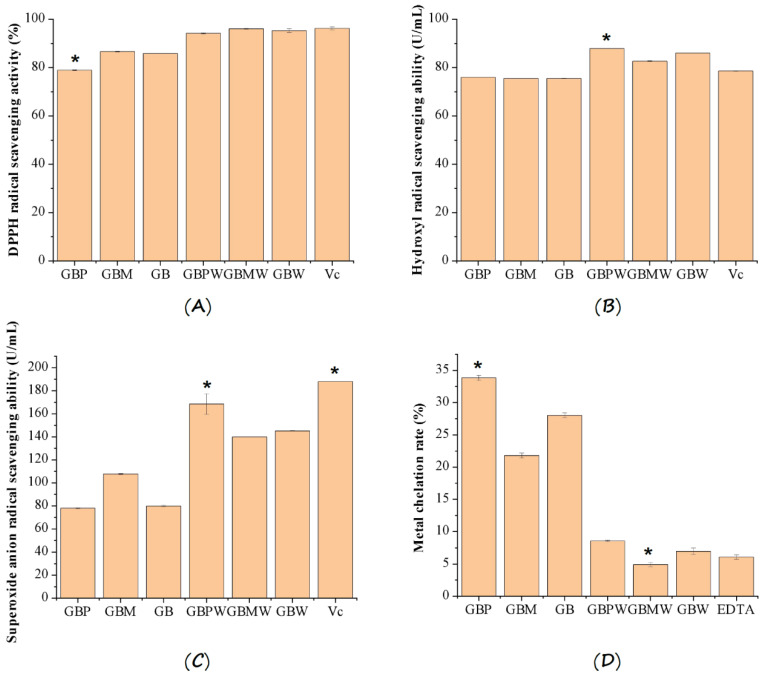
(**A**) DPPH radical scavenging activity, (**B**) hydroxyl radical scavenging ability, (**C**) superoxide anion radical scavenging activity, and (**D**) metal chelation rate of green banana and green banana wine samples. * denotes significant difference from other samples.

**Figure 5 antioxidants-12-00093-f005:**
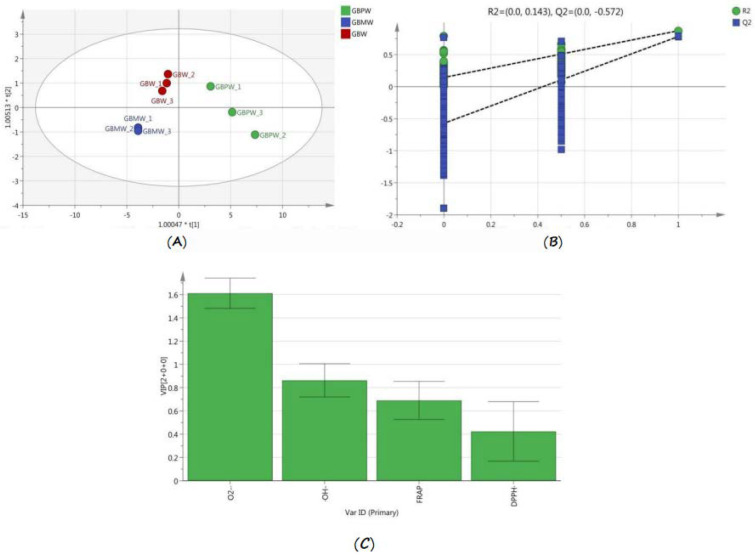
OPLS-DA model of antioxidant activities of green banana wine samples. (**A**) OPLS-DA score plot, (**B**) permutation tests for the OPLS-DA model, and (**C**) column plot of the variable average intensity of the four in vitro antioxidant activity assays.

**Figure 6 antioxidants-12-00093-f006:**
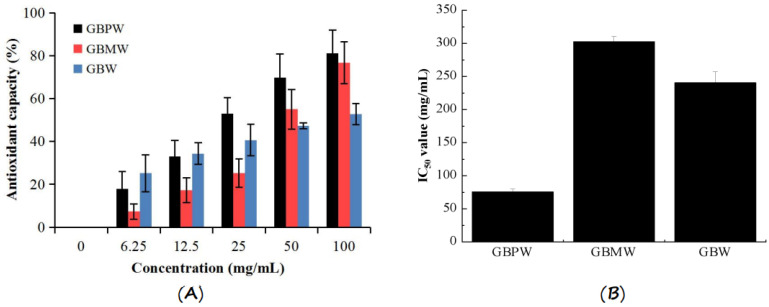
(**A**) Antioxidant capacities and (**B**) IC_50_ values of green banana wine samples on cell viability of 3T3-L1 preadipocytes.

**Table 1 antioxidants-12-00093-t001:** Mass spectrometric parameters of nine phenolic compounds.

Compounds	Retention Time (min)	[M + H]^+^ (*m*/*z*)	[M − H]^−^ (*m*/*z*)	MS/MS Fragments	Molecular Weight (g/mol)	Chemical Formula
Gallic acid	1.69	-	168.90	124.91, 78.90	170.12	C_7_H_6_O_5_
Gentisic acid	2.83	-	152.90	108.90, 80.92	154.12	C_7_H_6_O_4_
Protocatechuic acid	2.83	-	152.90	108.90, 90.88	154.12	C_7_H_6_O_4_
Chlorogenic acid	3.70	-	353.03	191.00, 84.88	354.31	C_16_H_18_O_9_
Catechin	3.82	291.07	-	138.96, 123.04	290.27	C_15_H_14_O_6_
Caffeic acid	4.55	-	178.90	134.93, 106.88	180.16	C_9_H_8_O_4_
Vanillic acid	4.57	-	166.9	151.93, 107.92	168.14	C_8_H_8_O_4_
Epicatechin	4.66	291.13	-	139.09, 123.10	290.27	C_15_H_14_O_6_
Syringic acid	4.75	-	196.97	181.96, 122.89	198.17	C_9_H_10_O_5_
p-Coumaric acid	5.99	165.11	163.0	119.07, 91.12 (ES+) 119.00, 93.30 (ES−)	164.16	C_9_H_8_O_3_
Ferulic acid	6.74	194.99	192.97	145.01, 116.99	194.18	C_10_H_10_O_4_
Salicylic acid	8.33	-	136.90	92.89, 64.91	138.12	C_7_H_6_O_3_
p-Hydroxybenzoic acid	8.33	-	136.91	92.89, 64.91	138.13	C_7_H_6_O_3_
Luteolin	8.37	287.10	-	153.07, 121.09	286.24	C_15_H_10_O_6_
Coumarin	8.58	147.07	-	91.12, 77.08	146.14	C_9_H_6_O_2_
Cinnamic acid	8.97	149.00	-	131.00 103.00	148.16	C_9_H_8_O_2_
Apigenin	9.15	271.10	-	153.07, 119.07	270.24	C_15_H_10_O_5_
Kaempferol	9.28	287.09	-	153.07, 121.09	28623	C_15_H_10_O_6_

**Table 2 antioxidants-12-00093-t002:** Phenolics content in green banana wine samples.

No	Phenolics	Concentration (μg/mL)			Minimum Detectable Concentration (μg/mL)
		Green Banana Peel Wine	Green Banana Pulp Wine	Green Banana Wine	
1	Gallic acid	4.536	0.46	1.209	4.59
2	Gentisic acid	0.009	0.038	ND	2.53
3	Protocatechuic acid	0.22	ND	ND	1.64
4	Chlorogenic acid	ND	ND	ND	8.38
5	Catechin	0.034	ND	0.007	5.42
6	Caffeic acid	0.782	0.048	0.011	9.27
7	Vanillic acid	ND	ND	ND	5.11
8	Epicatechin	0.109	0.124	0.131	2.03
9	Syringic acid	0.042	0.012	0.021	3.44
10	p-Coumaric acid	0.337	0.013	ND	4.35
11	Ferulic acid	10.493	0.047	0.286	15.49
12	Salicylic acid	0.065	ND	0.040	5.85
13	p-Hydroxybenzoic acid	0.055	ND	0.038	7.17
14	Luteolin	ND	ND	ND	3.19
15	Coumarin	ND	ND	ND	4.06
16	Cinnamic acid	ND	ND	ND	25.87
17	Apigenin	0.0013	ND	ND	0.46
18	Kaempferol	0.007	ND	0.005	2.22

“ND” indicates that the component was not detected.

**Table 3 antioxidants-12-00093-t003:** Flavor components in green banana peel (GBPW), pulp (GBMW), and whole fruit wine (GBW) samples.

No	Compounds	Relative Content (%)			Threshold (mg/L)	Characteristic
		GBPW	GBMW	GBW		
	*Fatty Acid Esters*					
1	Ethyl palmitate	1.40	3.57	2.03	2.26	Weak, waxy, and creamy aromas
2	Ethyl octanoate	-	0.66	-	0.17	Brandy aroma
3	Ethyl pelargonate	-	0.63	-	0.85 × 10 ^−3^	Cantaloupe flavor
4	Ethyl caprate	-	0.25	-	0.2	Coconut flavor
5	2,6-Diethylphenyl isocyanate	-	1.88	-	-	-
6	Ethyl laurate	-	1.72	1.61	1.5	Floral and fruity aromas
7	Ethyl tridecanoate	-	0.18	-	-	-
8	Ethyl undecanoate	-	0.84	-	1	-
9	Ethyl myristate	-	2.85	2.31	4	Iris aroma
10	13-Methyl-tetradecanoic acid ethyl ester	-	0.10	-	-	-
11	Ethyl pentadecanoate	-	0.67	-	-	-
12	Methyl palmitate	-	0.28	-	4	-
13	Ethyl stearate	-	0.16	-	-	Faint waxy scent
	*Acids*					
14	Acetic acid	-	0.98	-	-	Pungent smell
	*Alcohols*					
15	3-Methyl-1-butanol	72.02	54.04	76.49	0.3	Fruity and mellow flavors
16	2,3-Butanediol	4.52	0.70	-	-	-
17	(2R,3R)-(-)-2,3-Butanediol	-	4.79	-	-	-
18	4-Carvomenthenol	-	0.13	-	-	-
	*Alkenes*					
19	(+)-α-Longipinene	-	0.20	-	-	-
20	Longifolene	-	4.26	2.11	-	-
21	Bicyclo[7.2.0]undec-4-ene,4,11,11-trimethyl-8-methylene-,(1R,4Z,9S)-	-	0.58	-	-	-
	*Phenol*					
22	2,4-di-tert-butylphenol	19.06	9.16	11.53	-	Phenol odor
	**Total Content**	**97.00**	**88.63**	**96.08**		

“-” indicates that the compound was not detected.

**Table 4 antioxidants-12-00093-t004:** ROAV value of flavor substances in green banana peel (GBPW), pulp (GBMW), and whole fruit wine (GBW) samples.

No	Compounds	ROAV		
		GBPW	GBMW	GBW
	*Fatty Acid Esters*			
1	Ethyl palmitate	0.243	0.620	0.352
2	Ethyl octanoate	-	1.523	-
3	Ethyl pelargonate	-	-	-
4	Ethyl caprate	-	0.490	-
5	2,6-Diethylphenyl isocyanate	-	-	-
6	Ethyl laurate	-	0.450	0.421
7	Ethyl tridecanoate	-	-	-
8	Ethyl undecanoate	-	0.329	-
9	Ethyl myristate	-	0.279	0.227
10	13-Methyl-tetradecanoic acid ethyl ester	-	-	-
11	Ethyl pentadecanoate	-	-	-
12	Methyl palmitate	-	0.027	-
13	Ethyl stearate	-	-	-
	*Acids*			
14	Acetic acid	-	-	-
	*Alcohols*			
15	3-Methyl-1-butanol	94.156	70.650	100.000
16	2,3-Butanediol	-	-	-
17	(2R,3R)-(-)-2,3-Butanediol	-	-	-
18	4-Carvomenthenol	-	-	-
	*Alkenes*			
19	(+)-α-Longipinene	-	-	-
20	Longifolene	-	-	-
21	Bicyclo[7.2.0]undec-4-ene,4,11,11-trimethyl-8-methylene-,(1R,4Z,9S)-	-	-	-
	*Phenol*			
22	2,4-di-tert-butylphenol	-	-	-

“-” indicates that ROAV of the compound is not determined.

## Data Availability

All data are available in the article.

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
