# Peer review of "Development of Green Banana Fruit Wines: Chemical Compositions and In Vitro Antioxidative Activities"

_antioxidants, 2022, doi:10.3390/antiox12010093_

Round 1
Reviewer 1 Report
see attached file

Author Response
Respond to Reviewer 1’s comments:
Comment 1: Overall the paper presents interesting results for use of green banana materials in the production of fruit wines with the added benefit of finding an alternative use of waste materials from green bananas.
Response 1: Thank you for reviewing this manuscript.
Suggested revisions
Comment 2: Throughout the paper, the terminology of increasing or decreasing “trends” is used and not specific enough – please be more specific on the trend that is referred to and only uses it when necessary – in many instances, this sentence is not required.
Response 2: We have revised the sentences accordingly throughout this manuscript. The word “trend” has been replaced by “level”, “rate”, “value”, etc.
Comment 3: Line 47 specifies how waste is disposed of it is an environmental issue or removed "and also pollutes the environment" -this comment repeats in next paragraph and throughout this paper so is also repetitive with not enough detail on disposal methods typically used.
Response 3: We have revised the statements (page 2, lines 51-53). Banana waste is traditionally used as compost. Due to the advancement of science and technology, banana waste has been developed into a carbon-based adsorbent for wastewater treatment, fuel, and animal feed.
Comment 4: Lines 69-71 can delete “A large volume of waste is produced annually during banana production and processing because no special recycling treatment is available. Therefore, fully utilizing these wastes to improve the economic benefits of the banana industry is necessary and urgent.” This is repetitive in introduction – otherwise provide more detail on the normal waste disposal approach.
Response 4: We have revised the statement on page 3 (lines 81-82).
Comment 5: Line 75 banana industrialization- possible this should be recycling of banana wastes or alternative uses.
Response 5: We have revised the statement on page 3 (lines 87-88).
Comment 6: Line 110 The cone-hole gas (add in nitrogen)
Response 6: We have revised it accordingly (page 4, line 128).
Comment 7: Line 112. Using 18 phenolic standards at what calibration range? And what linearity was achieved – no information is provided in paper
Response 7: We added the information on page 4 (lines 130-133).
Comment 8: Line 120 inlet and connecting rod – are you referring to the interface temperature for the connecting rod?
Response 8: We have revised it as on page 4 (lines 143-144).
Comment 9: Line 122 Amu use amu.
Response 9: We have revised it as on page 4 (line 145).
Comment 10: Line 123 – no information on calibration range or linearity for standards in this paper – need to add additional information
Response 10: We performed the GCMS analysis and quantified the flavor components based on the relative odor activity value (ROAV). No pure standard was used in this GC analysis.
Comment 11: Line 140 and was a potential compound – not clear what you are referring to potential compound for what?
Response 11: We have removed the statement.
Comment 12: Line 149 respectiv should be respective
Response 12: We have revised it accordingly on page 5 (line 176).
Comment 13: Line 167 decreased with increasing fermentation time with the greatest reduction observed within 6 days, whereas alcohol content increased the most during this time or period, and then increased more gradually. After day 6 the decline of TSS was less than 1%.
Response 13: We have revised the sentences on page 6 (lines 207-210).
Comment 14: Line 187 had increasing trends -specify the trend not increasing or decreasing trend -rewrite this sentence or just remove it.
Response 14: We have revised these sentences on pages 6 and 7 (lines 239-244).
Comment 15: Line 214 – pH value rapidly decreases change – pH value to rapidly decrease.
Response 15: We have revised it accordingly on page 7 (line 254).
Comment 16: Line 218 The reduced concentration of organic acids led to less increase in pH led to a less increase should be reworded to a smaller increase or a more gradual increase.
Response 16: We have revised it accordingly on page 7 (lines 259-260).
Comment 17: Line 219-220 the most notable upward trend, - again specify the trend be specific about what you are referring to.
Response 17: We have revised it accordingly on page 7 (lines 260-261).
Comment 18: Line 269-270 an overall decreasing trend during fermentation -again be specific in describing the trend and variables that are changing.
Response 18: We have revised it accordingly on page 8 (line 306).
Comment 19: Line 324-325 - Also, the initial screening of phenolic compounds was done using HPLC. The results showed that gallic acid was not detected in banana wine samples. – What specifically is this HPLC method if it is not the UPLC?
Response 19: It should be the same UHPLC method. We have revised the statement accordingly on page 10 (lines 347-348).
Comment 20: MS method described in the experimental section. No other method with HPLC-detector? For screening was described.
Response 20: Identification and quantification of phenolic compounds were done using UHPLC coupled with an MS detector. For screening, we only used a UV-DAD detector. We have revised the statement accordingly on page 4 (lines 133-134).
Comment 21: What method was used for gallic acid?
Response 21: We used the same method with UV-DAD at the wavelength of 280 nm.
Comment 22: Table 1 Caffeic acid not completed all columns
Response 22: We have revised it as shown in Table 1 (page 10).
Comment 23: Line 34 abstract (as well as conclusions), the authors would benefit in discussing the feasibility of use of GBPW and GBW wine relative to GBMW given ethe ability to use green banana peel and thus find a viable usage of this waste material
Response 23: We have revised the statement on page 1 (Abstract, lines 38-39) and page 21 (Conclusions, lines 637-640).

Reviewer 2 Report
In this work, the authors developed several functional banana fruit alcoholic beverages using the whole fruit, banana pulp, and banana peel. The authors did a comprehensive analysis and obtained some interesting data. While there were many obvious errors that the authors may need to address and/or edit. The detailed comments were as follows:
Line 82, 3.5 mL or 3.5 L?
Line 84 What is the purpose of boiling processing for the 3.0 L banana mixture before enzymatic treatment?
Line 86, 0.5 g/L pectin or pectinase?
Line 150, For the cell culture-Based antioxidant capacity of banana wine, what is the positive and negative control samples? The detail information should be provided so that others can repeat your work by following your protocol in the near future.
How about the growth kinetics of yeast during fermentation? Please provide this information if available.
Lines 314 and 329, 9 phenolic compounds or 19 phenolic compounds?
In Table 3, please provide the linear retention index values of the characterized flavor compounds, it was not enough for the characterization of volatile by just comparing the spectrum with the Database.
Author Response
Respond to Reviewer 2’s comments:
Comment 1: In this work, the authors developed several functional banana fruit alcoholic beverages using the whole fruit, banana pulp, and banana peel. The authors did a comprehensive analysis and obtained some interesting data. While there were many obvious errors that the authors may need to address and/or edit. The detailed comments were as follows:
Response 1: Thank you for reviewing this manuscript. We have revised the manuscript accordingly.
Comment 2: Line 82, 3.5 mL or 3.5 L?
Response 2: It is 3.5 liter.
Comment 3: Line 84 What is the purpose of boiling processing for the 3.0 L banana mixture before enzymatic treatment?
Response 3: The leftover 3.0 L was boiled for 10 min to denature all enzymes and destroyed microbes. We have revised the statement on page 3 (lines 98-99).
Comment 4: Line 86, 0.5 g/L pectin or pectinase?
Response 4: It is pectinase. We have revised it on page 3 (line 100).
Comment 5: Line 150, For the cell culture-based antioxidant capacity of banana wine, what is the positive and negative control samples? The detail information should be provided so that others can repeat your work by following your protocol in the near future.
Response 5: We have revised it accordingly on page 5 (lines 186-191).
Comment 6: How about the growth kinetics of yeast during fermentation? Please provide this information if available.
Response 6: We did not measure the growth kinetics of yeast during the fermentation.
Comment 7: Lines 314 and 329, 9 phenolic compounds or 19 phenolic compounds?
Response 7: It should be 18 phenolic standards. We have revised it on page 9 (line 334).
Comment 8: In Table 3, please provide the linear retention index values of the characterized flavor compounds, it was not enough for the characterization of volatile by just comparing the spectrum with the Database.
Response 8: We performed the GCMS analysis and quantified the flavor components based on the relative odor activity value (ROAV). No pure standard was used in this GC analysis. The flavor components were identified by spectra matching using the GC/MS library. The accurate masses of the molecular ion adducts were also determined. We have also revised the statement on page 4 (lines 146-149).

Reviewer 3 Report
The aim of the paper was the development of green banana fruit wines and the determination of their chemical compositions and assessment of in vitro antioxidative activity.
Although the idea is interesting, the study is of preliminary nature.
Specifically:
There is no sensory evaluation of the new wines. Sensory assessment is of outmost importance for novel foods and beverages.
A Principal Component Analysis of the flavor compounds would be very helpful and probably highlight potential differences among the new products.
The Discussion section is very poor. I think it should be improved, highlighting the most important findings of the study.
Author Response
Respond to Reviewer 3’s comments:
Comment 1: The aim of the paper was the development of green banana fruit wines and the determination of their chemical compositions and assessment of in vitro antioxidative activity. Although the idea is interesting, the study is of preliminary nature.
Response 1: Thank you for reviewing this manuscript. Yes. This is the first and preliminary study on the development of a novel wine using different parts of green banana.
Comment 2: Specifically, there is no sensory evaluation of the new wines. Sensory assessment is of outmost importance for novel foods and beverages.
Response 2: The aim of this study is focused on the development of green banana wines, their chemical composition and in vitro antioxidant activities. Sensory evaluation has been suggested for future study (page 21, lines 639-640).
Comment 3: A Principal Component Analysis of the flavor compounds would be very helpful and probably highlight potential differences among the new products.
Response 3: We performed OPLS-DA instead of PCA. PCA cannot distinguish samples between groups well, but OPLS-DA can achieve effective separation. Moreover, the classification prediction models built by OPLS-DA can be further used to identify more sample categories, which cannot be achieved by the exploratory PCA method.
Comment 4: The Discussion section is very poor. I think it should be improved, highlighting the most important findings of the study.
Response 3: We have revised the discussion section (page 19).

Round 2
Reviewer 2 Report
The authors have addressed my comments. Can be accepted in its current version.
Author Response
Comment: The authors have addressed my comments. Can be accepted in its current version.
Response: Thank you for reviewing and accepting this manuscript to be published in Antioxidants (MDPI).
Reviewer 3 Report
Although the manuscript has been improved, the discussion still needs substantial improvement (my comment was not addressed). To my opinion, the authors should discuss the fact that their work is preliminary, no sensory tests were performed, provide insights of the necessary next steps and highlight the market potential of the products.
Author Response
Comment 1: Although the manuscript has been improved, the discussion still needs substantial improvement (my comment was not addressed).
Response 1: We have substantially improved the discussion section.
Comment 2: In my opinion, the authors should discuss the fact that their work is preliminary, no sensory tests were performed, provide insights of the necessary next steps and highlight the market potential of the products.
Response 2: In the discussion section, we have included a statement that this work is preliminary. We also mentioned that no sensory test was performed. Moreover, we provided hint for future study that GBPW is a potent functional drink for promoting healthy drinking lifestyle and preventing of metabolic diseases.
